# Amenity counts significantly improve water consumption predictions

**Damian Dailisan**[ID][⊗]*, **Marissa Liponhay**[ID][⊗], **Christian Alis**[ID], **Christopher Monterola***

Analytics, Computing, and Complex Systems Laboratory (ACCeSs@AIM), Asian Institute of Management, Makati City, National Capital Region, Philippines

⊗ These authors contributed equally to this work.
* ddailisan@aim.edu (DD); cmonterola@aim.edu (CM)

**Data Availability Statement:** All pre-processed data files are available from https://github.com/access-at-aim/amenity-water-consumption.

**Funding:** DD, ML, CA, and CM acknowledge the Department of Science and Technology (DOST) of

## Abstract

Anticipating the increase in water demand in an urban area requires us to properly understand daily human movement driven by population size, land use, and amenity types among others. Mobility data from phones can capture human movement, but not only is this hard to obtain, but it also does not tell where the population is going. Previous studies have shown that amenity types can be used to predict people's movement patterns; thus, we propose using crowd-sourced amenity data and other open data sources as reasonable proxies for human mobility. Here we present a framework for predicting water consumption in areas with established service water connections and generalize it to underserved areas. Our work used features such as geography, population, and domestic consumption ratio and compared the prediction performance of various machine learning algorithms. We used 44 months of monthly water consumption data from January 2018 to July 2021, aggregated across 1790 district metering areas (DMAs) in the east service zone of Metro Manila. Results show that amenity counts reduce the mean absolute error (MAE) of predictions by 1,440 $m^3$/month or as much as 5.73% compared to just using population and topology features. Predicted consumption during the pandemic also improved by as much as 1,447 $m^3$/month or nearly 16% compared to just using population and topology features. We find that Gradient Boosting Trees are the best models to handle the data and feature set used in this work. Finally, the developed model is robust to disruptions in human mobility, such as lockdowns, indicating that amenities are sufficient to predict water consumption.

## Introduction

Understanding and having an accurate prediction of water consumption dynamics are vital to form sustainable water systems management and develop efficient decision support systems and policies, which benefit both government and water concessionaires. Predicting water consumption helps daily operations, short-term and long-term planning, system maintenance, and business expansion. Accurate predictions of water consumption significantly reduce energy costs in pumping water and benefit the consumers with adequate volumes of water

the Philippines with Project No. 8419, 2020 under the Collaborative R&D to Leverage the Economy (CRADLE) Program (https://s4cp.dost.gov.ph/programs/cradle/). The funders had no role in study design, data collection, and analysis, decision to publish, or preparation of the manuscript.

**Competing interests:** The authors have declared that no competing interests exist.

supply at reasonable pressures. Failure to anticipate demand increases may result in supply shortages, such as the crisis that happened in 2019 in Metro Manila, Philippines [1].

Researches in predicting water consumption are often city- and time-specific where predictions are demonstrated on selected locations or district metering areas (DMA) of selected cities for a specific period [2–9]. Most works conducted short-term forecasts (hourly, daily, and weekly) to optimize pumping schedules and resolve disputes by detecting anomalous consumption. These approaches are feasible in areas with real-time water consumption monitoring, typically through automatic meter reading (AMR) devices [7] but may not be applicable in areas lacking such sensors.

Medium-term predictions (1–10yrs) are usually applied to account for the change in the number of consumers, while long-term forecasts (20–30yrs) are used to plan major changes in the water system [10]. Although these works attempt to provide solutions to water problems of areas with established water service connections, to our knowledge, there are no works on predicting water consumption for relatively new or underserved areas, which is of national interest in developing countries. Understanding the dynamism of water consumption per locality can be used as a guide for business expansion by the water concessionaires and as a policy guide for government regulation.

As a case in point, in the Philippines, only 54.1% of the population had water connections in 2020, with a high disparity in access between urban (73.8%) and rural areas (33.0%) [11]. In Metro Manila, about 92.6% of households have direct piped access to clean water. This discrepancy results from a similar policy applied to low-demand rural areas and high-demand urban regions. The Philippine government regulates the pricing of utilities with a profit cap of ∼11% per consumer, driving water utility companies to expand their coverage in high population density areas where they hope to reach the maximum earnings for the minimum infrastructure cost. However, demand is not merely a function of population density but also of economic activities within the area, the extent of which is empirically unresolved. Moreover, for a new or relatively new area to be serviced, the government must be sufficiently proactive and have a tool to work with water concessionaires to ensure that water is available in every home.

The Manila Water Company, Inc. (MWCI), a water concessionaire that serves the eastern side of the National Capital Region, estimates the future water consumption of a relatively new area as a product of the projected population of the area and the typical consumption in liters per capita per day (lpcd). In their case, the lack of lpcd consumption for the new area to be serviced is resolved by interpolating it with the consumption of areas perceived to have similar profiles. This idea holds but the challenge is understanding how the profile similarities lead to consumption behavior. The problem becomes more complex and dynamic in the presence of pandemics when business regulations, and economic activities in general, are linked to the extent to which a locality is compromised.

Predicting water consumption has been achieved using statistical and machine learning models [4, 10, 12–15], including deep learning algorithms [6, 16, 17]. Studies have demonstrated that machine learning models provide more accurate water consumption prediction than the traditional statistical approaches [4, 18]. Implementations of deep learning algorithms have also shown good performances in forecasting water demand [5, 16, 19] and forecasting dam water levels [20]. Moreover, previous works show that land use and amenities drastically impact the movement of people in a city and hence their daily consumption and activities [21–23]. Still, amenities have not been used explicitly for water consumption prediction.

In this work, we envision the use of Support Vector Regression (SVR) and Random Forest (RF) as these have been previously shown to be useful in forecasting water demand [5, 10, 24] from the perspective of time series forecasting. We also include another tree-based ensemble

model called Gradient Boosting Machine (GBM) with its implementations (most notably LightGBM) having shown superiority in the M5 competition in 2020 in terms of accuracy and uncertainty of forecasts [25]. Work applying GBMs to forecast dam water levels has also shown comparable uncertainty levels with deep learning algorithms [20].

Several parameters commonly used to predict water consumption include climate or weather data, socio-economic characteristics of consumers, urban design, and demographics [9, 26, 27]. City population is also a key driver of water demand [28], indicating the need to include dynamic factors that affect human consumption patterns, such as human mobility. While attempts to include GPS traces of human mobility have demonstrated an improvement in short-term water demand predictions [10], the availability of such data and preprocessing remains a challenge.

As such, an alternative to raw GPS traces as a proxy for human mobility would benefit water demand prediction. City growth has been linked to the emergence of urban land-use patterns [29], and consequently, various amenities. Our recent work shows that human mobility may depend on the amenities present in an area [23]. With the increasing rate of urban development, it is critical to include amenities and land use patterns that may affect water consumption. Studies that account for the presence of gardens and swimming pools and the number of apartments and bathrooms have demonstrated the use of amenities as predictors of consumption.

This work provides a framework using machine learning models to predict future water consumption of unknown areas as a function of human mobility. Specifically, we include the number of amenities (e.g., leisure places, offices, schools) and the ratio of domestic consumption as predictors. These quantities indirectly capture human mobility and serve as reasonable proxies. While GPS data best captures actual human movement, access to this data is difficult and does not tell where the population is going; thus, we use crowd-sourced amenity data from OpenStreetMap. We demonstrate the robustness of our model in its ability to predict water consumption during unexpected circumstances that affect human mobility, such as in the case of a pandemic.

## Methods

### Study area

Our study area consists of localities served by the Manila Water Company, Inc. (MWCI), which are the eastern part of Metro Manila and parts of Rizal province (Fig 1). The study area is divided into 1790 District Metering Areas (DMAs). For each DMA, historical water consumption was measured monthly and covered from January 2018 until July 2021. These areas serve an estimated resident population of around 7.5 million (based on $30 \times 30$ m$^2$ population data [30] and the DMA shapefiles) and covers 371 km$^2$.

### Data

This section details the different datasets used as features in this work. Table 1 shows all features with descriptions. The following subsections will discuss the details of these datasets. Links to raw and preprocessed data files are provided S1 Appendix.

**Water consumption.** Water consumption data were obtained from meter readings of MWCI customers, which include both residential and commercial accounts. Meters were read manually at a monthly frequency due to the lack of AMR devices. These readings are then aggregated at the DMA level and provided by MWCI. Since MWCI also tags their customer accounts as business or residential, they also provided us with the domestic and non-domestic consumption ratio aggregated at the DMA level. We split our data into two sets: one spanning

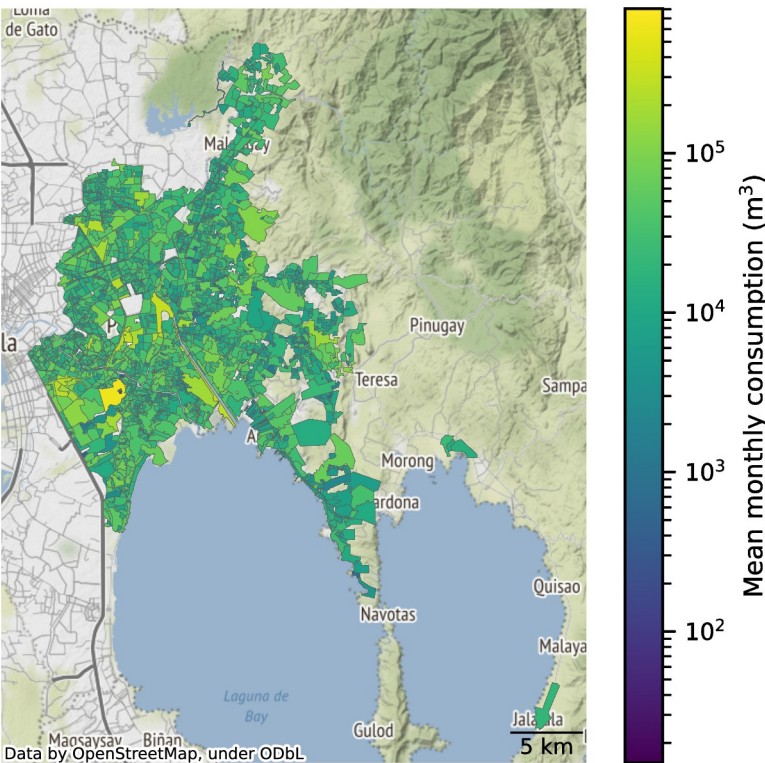

**Fig 1. Map of the study area.** Manila Water Company, Inc. services the eastern part of Metro Manila and portions of Rizal province. The DMAs cover 371 km$^2$ and cater to a resident population of around 7.5 million.

January 2018–11 March 2020 (pre-COVID), and the other spanning 12 March 2020–July 2021 (COVID). This split in data marks the period where the localities of Metro Manila and Rizal have been under some form of lockdown (with varying degrees of restriction) for the duration of the COVID dataset. We processed the water consumption data to exclude months with negative and zero consumption values. Negative values are typically the result of incorrect meter readings. Each month, agents manually record meter readings for billing purposes; errors, if any, are automatically discovered and corrected in the next month's billing. Distributions of retained monthly water consumption range from 15 to 806662 m$^3$ with a median of 14765.5 m$^3$, and is positively skewed (Fig 2). Filtering leaves a total of 76900 samples of a DMAs water consumption for a given month, with 46480 (60.4%) of the data under the pre-COVID period.

**Topology.**   We used Digital Elevation Model raster maps from the NASA Shuttle Radar Topographic Mission (SRTM) [31] that has a $30 \times 30$ m$^2$ spatial resolution. We then matched each pixel of the raster map with the DMA shapes, and extracted the mean, minimum, and maximum elevations within the DMA as features.

**Amenity data.**   We extracted OpenStreetMap (OSM) [32] data for amenity class counts in each of the different DMAs. Amenities are aggregated by groups (for amenity groups, see https://wiki.openstreetmap.org/wiki/Map_features) such as Sustenance, Accommodation, and Shop. An amenity may be classified under multiple groups. The full OSM history was used to generate monthly amenity counts over time. For example, to create an amenity count snapshot for January 2018, we filtered out all OSM elements added after January 31, 2018.

**Table 1. Description of all features used in this work.**

| Feature set | Feature | Description |
|---|---|---|
| Base | Elevation (m) | Elevation above sea level |
| | Population (2018) | Population aggregated within the DMA |
| | Area (m$^2$) | Land area of DMA |
| | Population Density (person/m$^2$) | Population divided by land area of DMA |
| | Year | Year of data sample |
| | Month | Month of data sample |
| Amenity counts | Accommodation | Buildings used as homes/hotels |
| | Civic/Amenity | Government offices and public infrastructure. It also includes schools and universities |
| | Commercial Discount store, charity | Used in any commercial activity, office buildings |
| | DIY | Do-it-yourself hardware, gardening stores, furniture, interior |
| | Education | Schools buildings |
| | Entertainment, Arts & Culture | Casinos, theaters, nightclubs |
| | Financial | Banks, ATMs, Money changers |
| | Health | Healthcare facilities, pharmacies, veterinary |
| | Landuse: Developed | Land where buildings may be placed |
| | Landuse: Rural and agricultural land | Land classified as rural or agricultural |
| | Landuse: Other | Other types of land use |
| | Leisure | Pools, resorts, spa, gyms, parks, sports stadia |
| | Religious | Places of worship |
| | Sustenance | Dining establishments, food stalls, pubs, bars |
| | Shops | Shopping malls, clothing, shoes, accessories |
| | Transport | Public transport terminals and stations, fuel stations, parking |
| | Others | Miscellaneous amenity classes |
| Domestic | % Domestic (0–100%) | The ratio of domestic to non-domestic consumption |
| Quarantine | [x] CQ days | Number of days in the month for the quarantine classification |

**Population data.** While census data would have been the ideal basis for population data, we faced the limitation that DMAs do not coincide with the resolution of the census (barangay-level, the smallest administrative division). As a workaround, we merged high-resolution population data and the DMA shapefiles to calculate the population inside each DMA (Fig 3). Population data was derived from Facebook's *Data for Good* High-Resolution Settlement Layer (HRSL), which has a spatial resolution of 30 × 30 m$^2$ [30]. We used the October 10, 2018 dataset for the population counts.

**Quarantine levels.** At the local onset of the COVID-19 pandemic in March 2020, the Philippines enforced a lockdown quarantine protocol called the Enhanced Community Quarantine (ECQ) to mitigate the spread of the virus. With ECQ in place, only essential establishments were allowed to open while only essential workers [33] were permitted to report to their workplaces. The national government subsequently implemented different types of quarantine protocols with varying restrictions depending on the severity of the number of cases in the country. The quarantine protocols used in this work include ECQ, Modified Enhanced Community Quarantine (MECQ), General Community Quarantine with heightened restrictions, and General Community Quarantine (GCQ), arranged according to decreasing restrictions of human mobility [33]. Collated data on the dates and durations of the enforced quarantine levels in our study area across various news releases are provided in S1 Appendix.

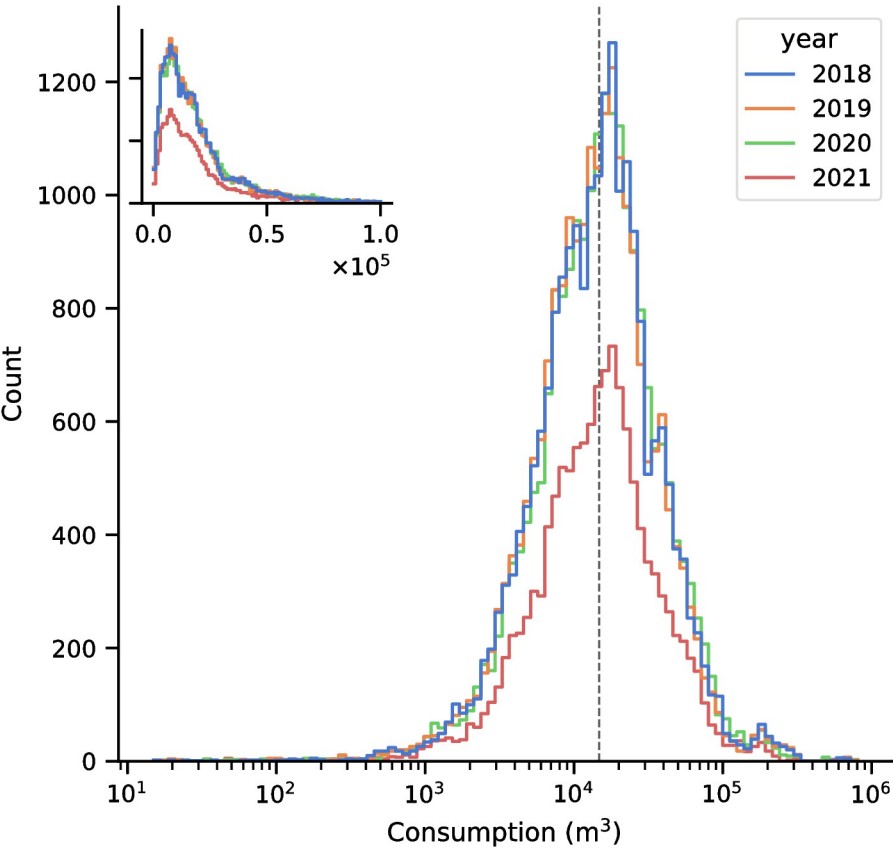

**Fig 2. Monthly consumption of the different DMAs.** The monthly consumption distributions of the 1790 DMAs range from 15 to 806662 m³ with a median of 14765.5 m³ (gray dashed line). The overall distribution has a positive skew of 7.67 and a kurtosis of 111.8. The inset shows the distribution on a non-logarithmic x-axis.

## Models

While deep learning methods have gained popularity in water consumption prediction applications, simple machine learning models avoid the complexities of deep learning models whilst providing acceptable performance.

**Tree-based ensemble methods.** These are non-parametric classes of supervised learning algorithms that employ a tree-like structure. Ensemble methods use large collections of trees to each predict a value or classification; the model then averages the results of each tree. While individual trees excel at learning patterns in data, they tend to create complex structures that do not generalize data well. Using an ensemble of trees provides good results while avoiding overfitting [34].

Random Forests (RFs) [35] use an ensemble of decision tree predictors $h_m(\mathbf{x_i})$, $m = 1, 2, \ldots, M$ trained on subsets of samples (bagging) of the data. The RF algorithm predicts a value

$$\hat{y}_i = F(x_i) = \frac{1}{M} \sum_{m=1}^{M} h_m(x_i), \tag{1}$$

A single decision tree [36] recursively partitions the feature space such that the samples $(\mathbf{x_i}, y_i)$ with similar values are grouped together. The algorithm for building a decision tree is as follows: let the data at node $k$ be represented as $Q_k$ with $N_k$ samples. For each candidate feature $x_j$,

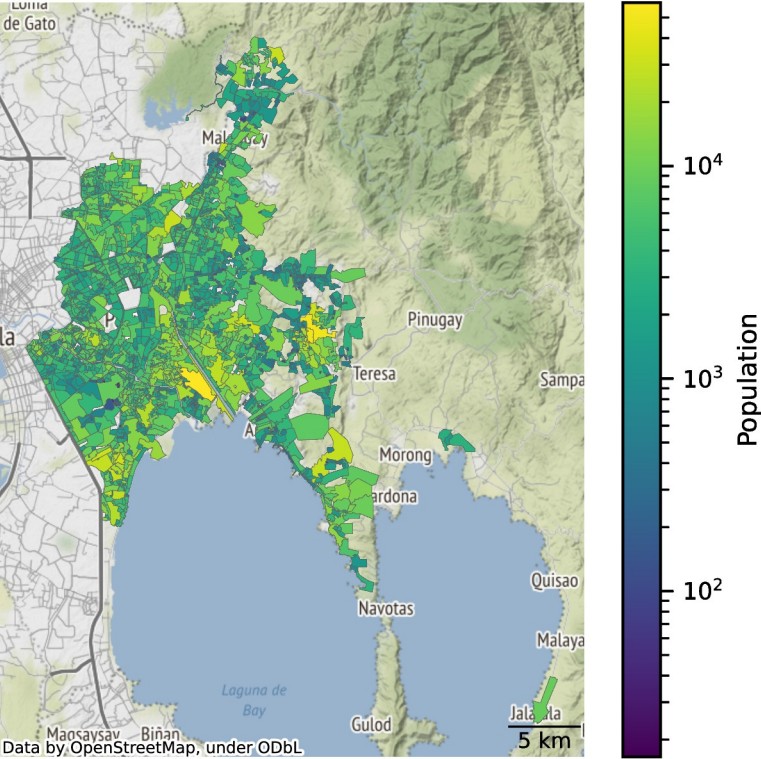

**Fig 3. Population data aggregated at the DMA level.** The DMAs cover 371 km$^2$ and cater to a resident population of around 7.5 million.

the data is split according to the threshold $t_k$ into $Q_k^{left}$ and $Q_k^{right}$ subsets

$$Q_k^{left} = \{(\mathbf{x_i}, y_i) | x_j \leq t_k\}, \tag{2}$$

$$Q_k^{right} = Q_k \backslash Q_k^{left}. \tag{3}$$

The quality of the split of the data is determined by the impurity metric $H(Q_k) = \frac{1}{N_k} \sum_{y \in Q_k} (y - \bar{y}_k)^2$, calculated over the set $Q_k$

$$G(Q_k) = \frac{N_k^{left}}{N_k} H(Q_k^{left}) + \frac{N_k^{right}}{N_k} H(Q_k^{right}). \tag{4}$$

The actual feature $x_j$ and split threshold $t_k$ are the values that minimize the impurity $G(Q_k)$. The above process is repeated until the *maximum depth* is reached or $N_k$ reaches the minimum number of samples. Random Forests train multiple trees in this fashion trained a random subset of the training set. Each split may be determined by using all input feature $Q_k$ or a random subset of size *max features*. The final prediction of RF averages each tree's predictions. By combining diverse trees, RF prevents overfitting, and bagging stabilizes model predictions through reduced variance. We used the `RandomForestRegressor` [37] implementation of scikit-learn with four tunable parameters: *max depth*, *max features*, *n estimators* (the number of trees), and *min samples leaf*.

Gradient Boosting Trees (GBTs) [38] employ the use of multiple decision trees in an ensemble, similar to RFs. Unlike RFs that grow trees in parallel, GBTs use weakly learning

decision trees, which are trained in sequence. GBTs are initialized to estimate the target variable using the average value For $M$ trees as estimators, the Gradient Boosting algorithm then iteratively constructs the model from $M$ decision trees as follows:

$$\hat{y}_i = F_M(x_i) = \sum_{m=1}^{M} \alpha h_m(x_i), \tag{5}$$

where $M$ is the number of trees, $\alpha$ is the learning rate, and $h_m(x_i)$ are the succeeding decision trees. Each tree $h_m(x_i)$ is fitted to minimize the loss function $L$ which is usually the mean squared error

$$L = \frac{1}{n_{samples}} (y - F_{m-1}(x))^2, \tag{6}$$

which gives us

$$h_m(x_i) = -\frac{\partial L}{\partial F_{m-1}} = \frac{2}{n_{samples}} (y - F_{m-1}(x_i)). \tag{7}$$

Combining weak learning trees in this fashion achieves improved accuracy and reduced variance provided appropriate fine-tuning of model parameters. Algorithms for GBTs differ in finding the best split points to learn the trees. Popular implementations of GBTs use pre-sorted bins (scikit-learn's GradientBoostingRegressor [37], XGBoost [39]) or histogram-based estimators (LightGBM [40], scikit-learn's HistGradientBoostingRegressor). We used GradientBoostingRegressor and LightGBM for this work and tuned our models for the following hyperparameters: *learning rate* (which controls the gradient update steps), *max depth*, and *n estimators*.

**Support Vector Regression.** Support Vector Machines (SVMs) [41] are supervised learning models for classification and regression. Support Vector Regression (SVR) [42] refers to SVMs used for regression problems. SVMs work by finding the hyperplane defined by **w** that maximally separates points in high-dimensional feature space. A generalized SVR takes the form [43]

$$\hat{y}_i = F(x_i) = \langle \mathbf{w}, \Phi(\mathbf{x_i}) \rangle - b, \tag{8}$$

where $\Phi$ denotes a non-linear transform. In SVR, the fitted hyperplane **w** minimizes the loss function,

$$\frac{1}{2} ||\mathbf{w}||^2 + C \sum_{i=1}^{n_{samples}} \max\left(0, |y_i - \langle \mathbf{w}, \Phi(\mathbf{x_i}) \rangle - b| - \epsilon\right), \tag{9}$$

where $C$ is a constant and $\epsilon$ is the tolerance margin. The hyperplane is then expressed as

$$\mathbf{w} = \sum_{i=1}^{n_{samples}} (\alpha_i - \alpha_i^*) \Phi(\mathbf{x_i}). \tag{10}$$

This allows us to rewrite the SVR equation as

$$\hat{y}_i = F(x_i) = \sum_{j=1}^{n_{samples}} (\alpha_j - \alpha_j^*) \langle \Phi(\mathbf{x_j}), \Phi(\mathbf{x_i}) \rangle - b \tag{11}$$

$$= \sum_{j=1}^{n_{samples}} (\alpha_j - \alpha_j^*) K(\mathbf{x_j}, \mathbf{x_i}) - b, \tag{12}$$

where the function $K(\mathbf{x_j}, \mathbf{x_i})$ is known as the kernel function. Non-linear kernels allow SVM to handle non-linear problems SVR can use a non-linear kernel to transform the data into a higher dimensional feature space; a linear hyperplane can then separate the data. We use the `SVR` implementation of scikit-learn, which has two tunable model parameters: the margin of tolerance $\epsilon$ and the choice of the kernel transformation, which can be one of linear ($\langle \mathbf{x_i}, \mathbf{x_j} \rangle$), polynomial ($(\langle \mathbf{x_i}, \mathbf{x_j} \rangle + r)^d$), Gaussian radial basis ($\exp(-\gamma ||\mathbf{x_i} - \mathbf{x_j}||^2)$), and sigmoidal functions ($\tanh(\kappa \langle \mathbf{x_i}, \mathbf{x_j} \rangle + c)$).

## Machine learning workflow

This section details the machine learning workflow used in this work. Each of the models in the *Models* section goes through three steps in our workflow. First, the input data undergoes *Feature Scaling and Encoding* as a preprocessing step. The data and model then undergo *Model Selection and Tuning*, which trains the model using the data while finding the optimal set of hyperparameters to produce the best results. To qualitatively compare models, we need *Scoring* metrics to give us an objective comparison of the model performance.

**Feature scaling and encoding.** Machine learning estimators commonly require data to have similar scales. Because we use features with different scales and ranges, we employed feature scaling on the different features in our dataset. This work uses three scaling transformations for each feature point $x_i$ to $x_i'$: a log-transform and min-max scaler, given by

$$x_i' = \log\ x_i, \tag{13}$$

$$x_i' = \frac{x_i - \min(\mathbf{x})}{\max(\mathbf{x}) - \min(\mathbf{x})} \tag{14}$$

respectively, and One-hot encoding (OHE) [44] for categorical data. For a random categorical value $\mathbf{s}$ with $n$ possible distinct values $s_1, s_2, \ldots, s_n$, the One-hot encoding of a particular value $s_i$ is a vector $\mathbf{v}$ where every component $\mathbf{v}_{i \neq j} = 0$ and $\mathbf{v}_{i=j} = 1$.

Water consumption was log-transformed as we had previously removed all negative and zero readings. Date features (year and month) were transformed using OHE, as their integer representations' increasing/decreasing progression of does not necessarily correlate with their impact on the model. All other input features were rescaled using min-max scalers.

**Model selection and tuning.** For the given data period, we used $k$-fold group cross-validation with $k = 5$ to fine-tune each model's parameters. Cross-validation is a resampling method used in prediction problems to evaluate the generalization performance of a model and estimate the uncertainty of error statistics. In $k$-fold group cross-validation, the learning and training data are divided into $k$ sets of data (folds). One of the folds is then used as a testing set and the model is trained on the remaining $k - 1$ folds. This is repeated for each of the $k$ folds in the data. Data points were grouped by DMA labels, and the folds represent groups of DMAs. A DMA cannot be a part of more than one fold. We perform a grid search over possible combinations of tunable hyperparameters of the data. Each set of hyperparameters is scored using the $R^2$ metric (see the Scoring section), and the hyperparameter set that obtains the best score defines the model.

**Scoring.** We evaluate the results using four prediction accuracy measures: mean absolute error (MAE), mean absolute percentage error (MAPE), the coefficient of determination ($R^2$), and the Kling-Gupta Efficiency (KGE) [45]. Using the notation $\hat{y}_i$ to denote the predicted water consumption and $y_i$ as the actual consumption for each sample $i$, the measures are defined as follows:

$$\text{MAE}(y, \hat{y}) = \frac{1}{n_{\text{samples}}} \sum_{i=1}^{n_{\text{samples}}} |y_i - \hat{y}_i|, \tag{15}$$

$$\text{MAPE}(y, \hat{y}) = \frac{1}{n_{\text{samples}}} \sum_{i=1}^{n_{\text{samples}}} \frac{|y_i - \hat{y}_i|}{\max(\epsilon, y_i)}, \tag{16}$$

$$R^2(y, \hat{y}) = 1 - \frac{\sum_{i=1}^n (y_i - \hat{y}_i)^2}{\sum_{i=1}^n (y_i - \bar{y})^2}, \tag{17}$$

$$\text{KGE}(y, \hat{y}) = 1 - \sqrt{(r-1)^2 + \left(\frac{\sigma_{\hat{y}}}{\sigma_y} - 1\right)^2 + \left(\frac{\bar{\hat{y}}}{\bar{y}} - 1\right)^2} \tag{18}$$

where $\bar{y} = \frac{1}{n}\sum_{i=1}^n y_i$ denotes the mean water consumption and $\sigma$ is the standard deviation.

The MAE and MAPE both represent deviations of predictions from their actual values; MAE and MAPE scores of zero indicate perfect predictions. Because of the positive skew of the water consumption data, the MAE may be pulled up by larger errors for higher consumption DMAs. MAPE being a relative error metric allows us to compare errors across the extensive range of water consumption values. The $R^2$ value expresses the goodness-of-fit of the model, and ranges from $(-\infty, 1]$, with $R^2 = 1$ indicating a perfect fit and $R^2 = 0$ indicating that the model prediction is the mean ($\hat{y}_i = \bar{y}$). The KGE is a metric commonly used in evaluating hydrology models [46–48] that combines three statistical characteristics: Pearson's correlation $r$, variability bias, and mean bias. KGE values range from $(-\infty, 1]$ with KGE = 1 indicating a perfect fit. While no specific meaning is associated with KGE = 0, Knoben et al. [48] argues that values of KGE $> 1 - \sqrt{2}$ indicate that the model prediction is better than the mean, even if the KGE value is negative.

Using the chosen hyperparameters for each model, we employed another round of $k$-fold group cross-validation to evaluate the model performance on our dataset. For each iteration of cross validation, $k - 1$ folds was used to fit the model, and the remaining fold was used as a test set and generated the four scoring metrics. This was done for all $k = 5$ folds, and we then aggregated the results on each of the test data and reported the mean and standard deviation of the scores.

## Results

This section is divided into the three highlights of this work. First, we present an exploratory data analyses on representative DMAs and examine the changes in consumption patterns due to the COVID pandemic. We then present our model for predicting water consumption of DMAs using machine learning. Lastly, we look at the robustness of our model in predicting water consumption when population movement is reduced with the inclusion of pandemic restrictions as features.

### Analysis of changes to consumption patterns

We show the effects of community quarantines in three representative DMAs: commercial, residential, and mixed residential (50%) and non-residential (50%). We calculate the actual water consumption per month for every quarantine protocol using a linear model. Here we used the pre-pandemic (January 2018–11 March 2020) and pandemic (12 March 2020–July 2021) monthly consumption data for eastern Metro Manila and Rizal.

Quarantine measures during a pandemic disrupt human mobility and affect the water consumption patterns in residential, commercial, and industrial areas. The quarantine restricted

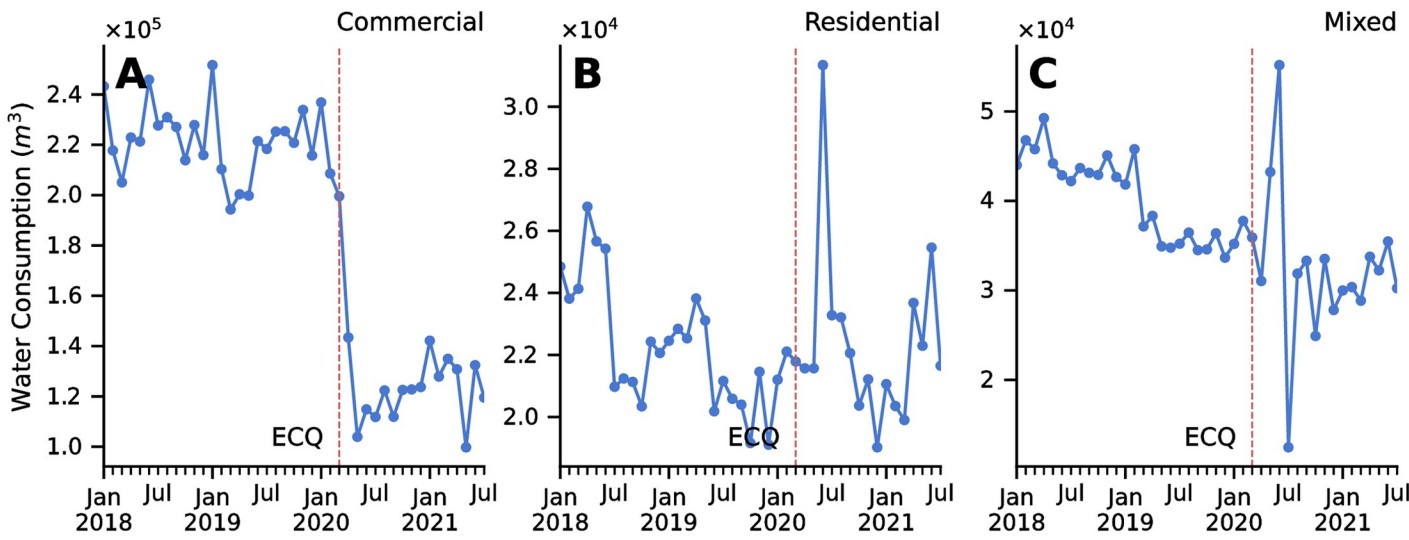

**Fig 4. Changes in water consumption due to quarantine restrictions.** Sample DMAs for (A) commercial areas, (B) residential areas, and (C) mixed residential and non-residential areas. Red dashed lines indicate Enhanced Community Quarantine (ECQ) was implemented in Metro Manila and Rizal.

movement and forced the majority of the population to stay at home. Consequently, patterns of water consumption changed in almost all DMAs. Fig 4A and 4B show the historical water consumption patterns of two representative DMAs: a commercial and a residential area, respectively. We can see that after the government implemented strict Enhanced Community Quarantine (ECQ) mid-March 2020, water consumption in the commercial area significantly decreased in the succeeding months while the consumption in the residential and in a DMA with mixed residential and non-residential areas both increased.

Roughly every two weeks, a government task force evaluates community quarantine protocols and recommends to modifying or retaining quarantine levels. Emergencies that cause a rapid surge of COVID cases may warrant an early update of quarantine protocols; stricter community quarantines were imposed for a few days to a week depending on the severity of infections in an area, such as the response to the delta variant surge. The very dynamic imposition of community quarantine levels resulted in possible combinations of two or more quarantine types in one month. The coupled consumption of different quarantine types in a single month poses a challenge in determining the actual water consumption attributed to a community quarantine type.

To decompose the monthly water consumption into individual consumption rates attributed to a type of community quarantine, we use a linear model to approximate the actual monthly consumption as a weighted sum of consumptions of the quarantine types imposed in a month:

$$m_i = \sum_q d_{i,q} n_q, \tag{19}$$

$$d_{i,q} = w_q \bar{m}_i, \tag{20}$$

$$m_i = \sum_q w_q \bar{m}_i n_q \tag{21}$$

Here, $m_i$ is monthly consumption, $d_{i,q}$ is monthly consumption rate for quarantine type $q$, $n_q$ is the fraction of days in the month of a quarantine type, and $w_q$ is the consumption weight. The

**Table 2. Consumption weights associated with quarantine types.**

|  | weights |
|---|---|
| ECQ | 0.849919 |
| MECQ | 0.819617 |
| GCQ | 0.808687 |
| GCQr | 0.829934 |
| Normal | 1.029549 |

$w_q$ coefficients are determined by fitting the COVID data to a linear regression model, and are presented in Table 2. We also accounted for the differences in quarantine types imposed on each DMA. A baseline of mean monthly consumption $\bar{m}_i$ six months prior to the COVID data (Sept. 2019–Feb. 2020) was used to fit the data.

Using these coefficients, we decomposed months with $k$ quarantine types into $k$ monthly consumption rates:

$$d_{i,q} = \frac{w_q m_i}{\sum_q w_q n_q} \qquad (22)$$

The normalization of Eq 22 ensures that the contributions $d_{i,q}$ of each quarantine type satisfies Eq 19.

Fig 5 shows the relative changes in water consumption per quarantine type with respect to the baseline consumption $\bar{m}_i$, of the commercial and residential DMAs, respectively. While the worst decrease in water consumption in the commercial DMA happened two months after ECQ was imposed, its consumption stabilized to approximately 53% of its pre-pandemic consumption on average for all the other levels of community quarantine. On the other hand, while the consumption in the residential DMA had the highest increase of $\sim 1.3$ times the

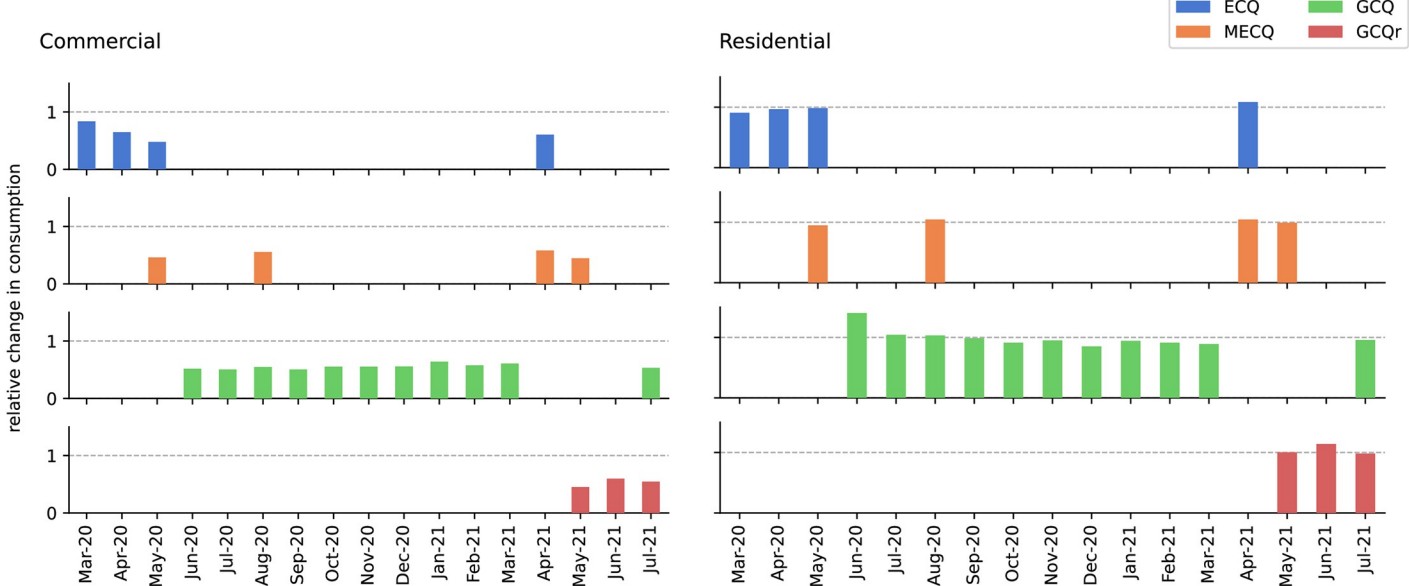

**Fig 5. Relative consumption of representative DMAs.** The left graph shows a sample commercial DMA, while the right shows a residential DMA. The commercial DMA has lower consumption than its pre-COVID baseline, while the residential DMA peaks around 3 months after the ECQ then drops down to near pre-COVID levels.

average pre-pandemic consumption two months after ECQ is declared, the consumption stabilized at approximately 96% of the pre-pandemic consumption for all the other levels of community quarantine. The increase in residential water consumption at the beginning of COVID-19 hints to people staying at home even on weekdays or more frequent washing and sanitation, while the unexpected decrease in water consumption as the quarantine progressed may hint to a few other behavioral changes while staying at home. Given these observed trends in consumption, we deem that quarantine protocols may be useful predictors for water consumption during a pandemic.

**Predicting consumption of DMAs.** A high percentage of the Philippine population still lacks piped water connections to their homes. Since government regulation caps profits from utilities, water utility companies have a strong incentive to expand their coverage to generate more revenue. Water consumption predictions for a new DMA can help potential water utility companies plan additional water lines to serve these areas while minimizing potential losses due to mismatched supply and demand.

Our machine learning methodology uses publicly available datasets as predictors to demonstrate an approach to predict average monthly consumption for a DMA. Because a new DMA will likely lack historical data on water consumption, we excluded this as a feature on our models. Since the amenity counts vary much slower than the fluctuations in consumption over the year, we removed the temporal aspect of the data and predicted the annualized average monthly consumption of DMAs. Only the pre-COVID part of the dataset was used in this analysis.

Table 3 shows the performance of our models in predicting DMA average consumption. The base set only used population and topological features (elevation and DMA area) as features (a complete list of features is shown in Table 1). We denote the use of amenity features as $A$ and percent domestic consumption as $D$. Using only the base set of features for predicting DMA consumption yields high errors, with MAE $\sim 11{,}000$ and the lowest $R^2$ values. Using amenity features alone yields similar MAE values across all models, but with worse MAPE

**Table 3. Evaluation metrics for predicted consumption of DMAs.**

| statistic | model | base | A | base + A | base + D | base + (A+D) |
|---|---|---|---|---|---|---|
| MAE | GradientBoosting | 11,170 (1,117) | 11,697 (882) | 9,729 (818) | 10,401 (519) | **9,182 (678)** |
| | LGBM | **10,913 (1,182)** | 11,846 (917) | 9,994 (831) | **10,210 (660)** | 9,232 (626) |
| | Random Forest | 10,957 (1,208) | **11,544 (885)** | **9,688 (783)** | 10,376 (618) | 9,278 (576) |
| | SVR | 11,142 (1,161) | 11,807 (1,022) | 10,300 (939) | 10,923 (880) | 9,932 (628) |
| MAPE | GradientBoosting | 76.79 (10.88) | 104.21 (29.55) | 71.06 (9.45) | 70.89 (15.52) | 62.90 (11.09) |
| | LGBM | **71.05 (5.75)** | 106.13 (30.39) | **69.33 (8.02)** | **62.41 (4.77)** | **56.68 (3.78)** |
| | Random Forest | 73.95 (8.46) | **103.92 (29.30)** | 71.04 (8.52) | 69.93 (11.10) | 62.08 (8.20) |
| | SVR | 88.58 (23.43) | 114.91 (37.70) | 84.77 (24.24) | 91.90 (32.63) | 76.05 (17.77) |
| $R^2$ | GradientBoosting | 0.48 | 0.35 | 0.56 | 0.54 | 0.61 |
| | LGBM | **0.51** | 0.35 | **0.57** | **0.57** | **0.62** |
| | Random Forest | 0.51 | **0.36** | 0.56 | 0.54 | 0.61 |
| | SVR | 0.46 | 0.34 | 0.52 | 0.44 | 0.51 |
| KGE | GradientBoosting | **0.25 (0.09)** | **0.44 (0.13)** | **0.48 (0.15)** | 0.30 (0.11) | 0.48 (0.09) |
| | LGBM | 0.24 (0.11) | 0.41 (0.10) | 0.47 (0.12) | **0.35 (0.08)** | **0.56 (0.11)** |
| | Random Forest | 0.20 (0.10) | 0.37 (0.11) | 0.47 (0.09) | 0.28 (0.07) | 0.46 (0.07) |
| | SVR | 0.20 (0.12) | 0.22 (0.09) | 0.35 (0.14) | 0.29 (0.09) | 0.45 (0.13) |

Reported scores are mean (stdev) values obtained from 5-fold group cross validation. Best models for each metric and feature set are highlighted in boldface.

(above 100%) and $R^2$ values. However, using the base set with amenity features improved the prediction of ensemble tree methods by as much as 1,441 m$^3$ (12.9% drop compared to MAE using the base set). The KGE metric further supports this observation, showing amenities clearly outperform the base set, with the correlation, mean, and variance taken into account. The best performing model is the GBT implementation of scikit-learn, which saw a 5.73% reduction in MAPE compared to just using the base features. These sets of features can be applied to the task of predicting potential new DMAs in urbanized areas (where amenity data from OSM is potentially available). Additional data collection or use of other sources for amenity type data can also extend this approach to completely new DMAs in rural areas.

We also explored using the ratio of domestic to non-domestic consumption as a feature. This addition further improved the prediction accuracy of all models, with the GBT yielding the best MAE score of 9,182 m$^3$. The MAPE was also further reduced by up to 14.37% for the LightGBM implementation of GBTs. All metrics point to the case of base + (A+D) as the best prediction model. However, the ratio of domestic to non-domestic consumption comes from MWCI's classification of customer accounts, something that might not be available when predicting consumption of currently unserved or underserved areas.

Large errors in our results shown in Table 3 may be attributed to the positive skew and a long tail of the consumption data. While the model was able to achieve $R^2$ values of 0.61 for log-transformed target variables, much of the errors come from the tail end of the data (Fig 6).

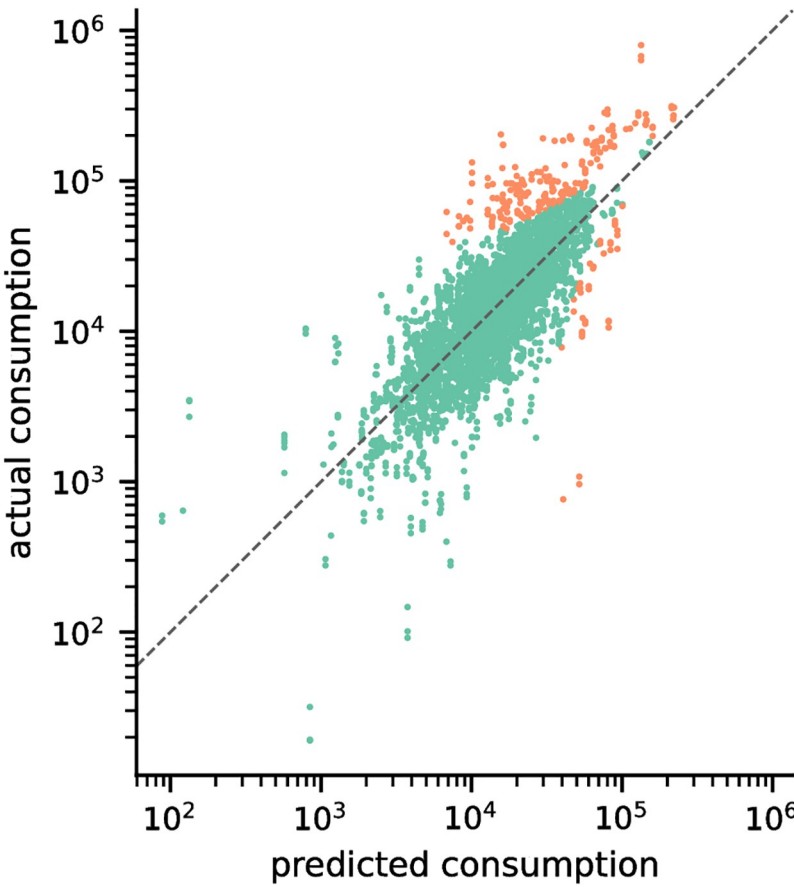

**Fig 6. Correlation plot of predicted and actual values.** Predicted vs. actual values obtained using the scikit-learn implementation of Gradient Boosting Trees using all features. The top 5% of the most significant residual errors are highlighted in orange.

Filtering out actual consumption values beyond the 95$^{th}$ percentile of the data, we calculate improved MAE scores of 6,496 m$^3$.

To gain more intuition on how the feature variables affect water consumption, we can look at the SHapley Additive exPlanations (SHAP) [49] feature importance scores of each feature (Fig 7). SHAP uses a game-theoretic approach to measure the contribution of each feature on the outputs of any machine learning model. Of the base features, we found that area and population are the most essential features in determining consumption. Because water consumption is aggregated at the DMA level, it is expected that larger or more populated areas would also have higher water consumption. The percentage of domestic consumption is also essential and, if available, should be included as a feature for our models. Likewise, the minimum and maximum elevations within a DMA may also influence water consumption; elevated areas are usually the most affected when service interruptions occur.

More interestingly, feature importance scores can help us determine the amenity types that contribute the most to predict water consumption accurately. The top three important amenity classes are *Leisure*, *Civic/Amenity*, and *Sustenance*. *Leisure* includes amenities such as water parks, swimming pools, parks, gardens, and sports stadia, all of which consume large amounts of water as part of operations. *Civic/Amenity* includes public buildings, government offices, transport infrastructure, fire stations, and educational institutions, which may be indicators of dense urban activity, which correlates with water consumption. Finally, amenities related to

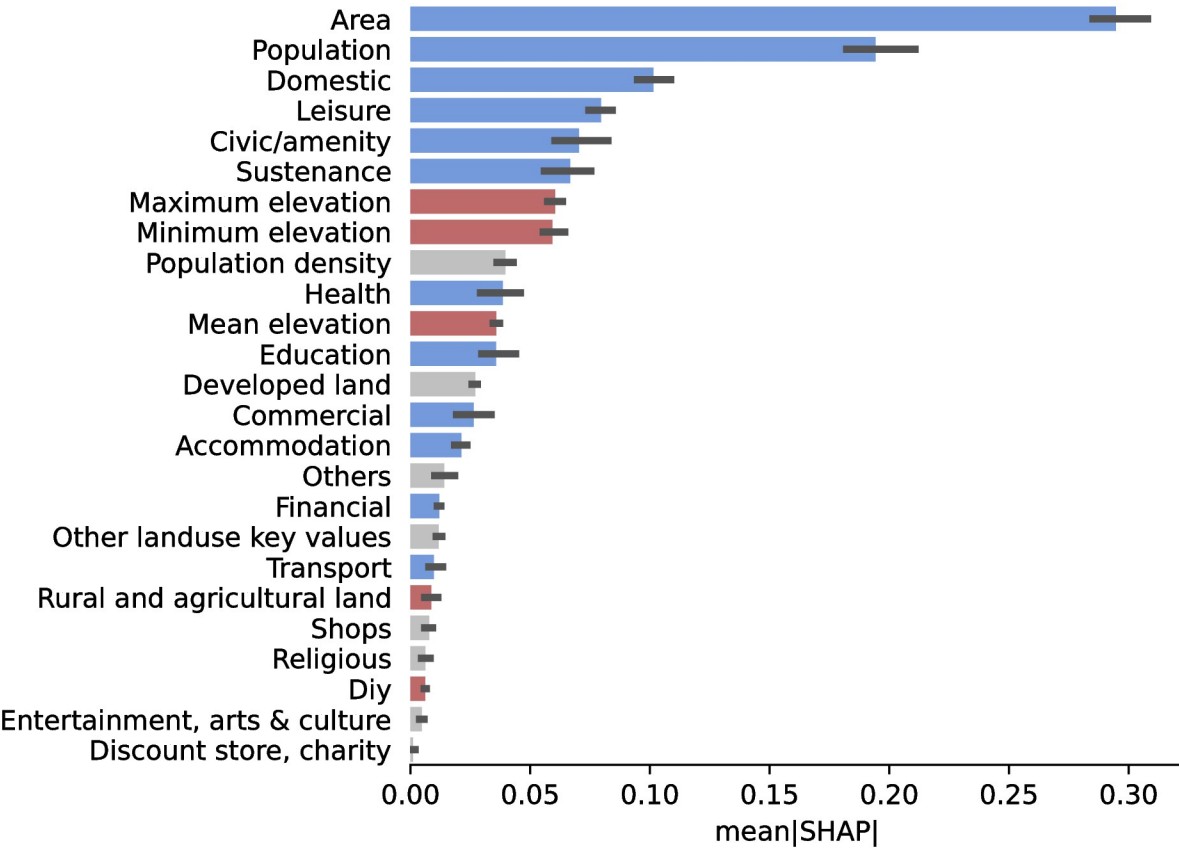

**Fig 7. SHAP Feature importance for a scikit-learn Gradient Boosting Tree.** Blue bars denote clear positive correlation, red for negative correlation, and gray bars have no clear correlation. Land area and population are the top two features that determine model predictions. Among amenities, *Leisure*, *Civic/Amenity*, and *Sustenance* are the top 3 predictive features.

*Sustenance* include restaurants, fast-food joints, and cafes. Apart from Leisure, these establishments serve as robust attractors to human activity, making them good predictors of an area's water consumption. On the other hand, establishments with low SHAP values are more niche to an area, with a couple of notable exceptions. Transport establishments have a strongly correlate with human movement, but these may not serve as good predictors for consumption as these may capture the transient aspect of mobility. Accommodation includes residential dwellings and, while having the highest amenity count, did not have high feature importance.

Current OSM data quality limits us to use amenity counts as our feature. As such, this measure naturally fails to capture differences between amenities of the same class—a larger mall probably has a different consumption than a smaller mall, but both will be counted equally as one—and may be improved by using the gross floor areas. However, while some buildings in OSM have polygonal shapes, which allow one to measure the floor area, building outlines will fail to capture the total floor area of multi-story buildings. We recommend extracting floor areas instead of just raw counts for each amenity type if the data allows for this.

## COVID-19 pandemic disruptions and quarantine feature labels

Given our results above, the presence of urban amenities and the ratio of domestic consumption of the DMAs are highly relevant predictors that indirectly captures human mobility. We demonstrate the ability of our models to learn the consumption patterns in the presence of disruptions in human mobility such as the COVID-19 pandemic. The entire training dataset (pre-COVID and COVID) is used without aggregating the monthly consumption. In addition to features used in the previous sections, we added parameters associated with the pandemic such as the number of days for each quarantine level and the year and month and apply the same feature selection for our machine learning models. Our results were then compared to our calculated base predictions with only population, elevation, area, and year and month as predictors.

Shown in Table 4 are the evaluation metrics of the different machine learning models used to predict DMA consumption with the COVID-19 pandemic taken into consideration. Similar

**Table 4. Evaluation metrics for predicted consumption of DMAs with COVID-19 pandemic disruptions.**

| statistic | model | base | base + A | base + (A+D) | base + (A+Q) | base + (A+Q+D) |
|---|---|---|---|---|---|---|
| MAE | GBR | **10,611 (1,098)** | 9,697 (1,150) | 9,360 (1,173) | 9,798 (1,003) | 9,369 (1,223) |
| | LGBM | 10,670 (1,114) | **9,554 (1,147)** | **9,126 (1,064)** | **9,542 (1,160)** | **9,228 (983)** |
| | RF | 10,996 (1,004) | 9,818 (1,003) | 9,580 (1,058) | 9,729 (1,031) | 9,573 (1,028) |
| | SVR | 11,386 (1,222) | 10,376 (1,374) | 10,508 (1,193) | 10,482 (1,348) | 10,638 (1,205) |
| MAPE | GBR | **75.54 (12.87)** | **68.70 (8.65)** | 65.49 (8.71) | 70.99 (11.81) | **63.60 (7.13)** |
| | LGBM | 80.84 (17.99) | 70.83 (9.87) | **63.78 (8.22)** | **70.37 (9.55)** | 65.31 (8.65) |
| | RF | 82.67 (14.78) | 73.13 (10.68) | 70.59 (9.98) | 72.48 (10.44) | 69.60 (9.40) |
| | SVR | 109.60 (35.84) | 100.99 (33.01) | 86.01 (23.00) | 102.55 (33.88) | 87.16 (23.58) |
| $R^2$ | GBR | **0.51** | **0.57** | 0.58 | 0.56 | **0.60** |
| | LGBM | 0.50 | **0.57** | **0.59** | **0.57** | 0.59 |
| | RF | 0.47 | 0.55 | 0.57 | 0.55 | 0.57 |
| | SVR | 0.37 | 0.44 | 0.46 | 0.43 | 0.45 |
| KGE | GBR | 0.23 (0.13) | 0.46 (0.16) | **0.55 (0.18)** | 0.46 (0.14) | 0.47 (0.16) |
| | LGBM | 0.25 (0.14) | 0.47 (0.17) | 0.51 (0.16) | 0.47 (0.16) | **0.55 (0.17)** |
| | RF | 0.27 (0.11) | 0.44 (0.11) | 0.45 (0.11) | 0.45 (0.10) | 0.46 (0.10) |
| | SVR | **0.29 (0.11)** | **0.50 (0.26)** | 0.48 (0.24) | **0.50 (0.26)** | 0.47 (0.23) |

Reported scores are mean (stdev) values obtained from 5-fold group cross validation. Best models for each metric and feature set are highlighted in boldface.

to our results in predicting DMA consumption above, the predictions have improved when we add the amenity counts *A* and the ratio of domestic consumption *D* as features (base + (A +D)). Further addition of quarantine-related factor *Q* (base + (A+Q+D)), resulted in 14% and 16% improvement in MAE and MAPE compared to base prediction, respectively; with LightGBM as the best model based on MAE and $R^2$ metrics. With the inclusion of amenities as features, even with the disruption due to the pandemic, implementations of GBR and LGBM perform the best among the MAE, MAPE, $R^2$, and KGE metrics. Although these improvements are significant when compared to the base prediction, the best models performed worse in terms of MAE and only slightly better (less than 1%) in terms of MAPE in the absence of *Q* (base + (A+D)). Thus, the amenities and ratio of domestic to non-domestic consumption are already sufficient predictors of water consumption even in the presence of pandemic disruptions.

To verify the importance of quarantine-related factors, we extract the feature importances of LightGBM as shown in Fig 8. Indeed, the quarantine-related features (*ECQ_days*, *MECQ_-days*, *GCQ_days*) have negligible importance compared to amenities and domestic consumption ratio. This result indicates that amenities capture long-term effects of human mobility and that quarantine effects are transient. Quarantines have little effect on water consumption in the long run, as the changes in water consumption due to quarantines are already reflected by the amenities that are responsible for water consumption.

Comparing Fig 8 to our results without pandemic disruption, *Civic/Amenity* drops in importance in line with *Health*. *Accommodation* moves up from 8th to 6th place among amenity types, overtaking *Education* which dropped from 5th to 7th. The increase in the importance of *Accommodation* reflects the shift in consumption to residential areas as quarantine restrictions limited mobility, while the decrease in the importance of *Civic/amenity*, *Leisure*, and *Education* reflects the closures of government and public buildings, areas for leisure activities, and educational institutions. We demonstrate that the use of amenity and ratio of domestic consumption allowed our model to successfully predict water consumption while being robust to the effects of a sudden disruption in human movements by requiring people to stay at home, among others. The robustness of our models stems from different amenity types capturing different usage patterns associated with different places that translate to human activities. Amenities reflect long-term patterns of human activity and changes in consumption indicate transient effects of disruptions on human mobility, such as quarantines.

## Conclusion

This work explored the feasibility of incorporating amenity counts as an additional feature to enrich machine learning predictions of water consumption data. We used three classes of machine learning models: Random Forest, Gradient Boosting Trees, and Support Vector Regression, to predict water consumption given combinations of population and topology, amenity counts, and the ratio of domestic to non-domestic consumption as features. Gradient Boosting trees performed best for these tasks, achieving the lowest mean absolute error of 9,182 m$^3$ when using all feature sets compared to an 11,170 m$^3$ error (17.8% increase in error) when using only baseline population and topology features. Amenity counts already accounted for more than half of this decrease in error and suggest the possible application of this method to predict water consumption in relatively underserved DMAs. Countries with low population percentages with piped water connection rates will benefit from such an analysis, making it easier to plan and manage the needs of new water consumers.

We also demonstrated the robustness of our model in its ability to predict water consumption during unexpected circumstances that affect human mobility, such as in the case of a

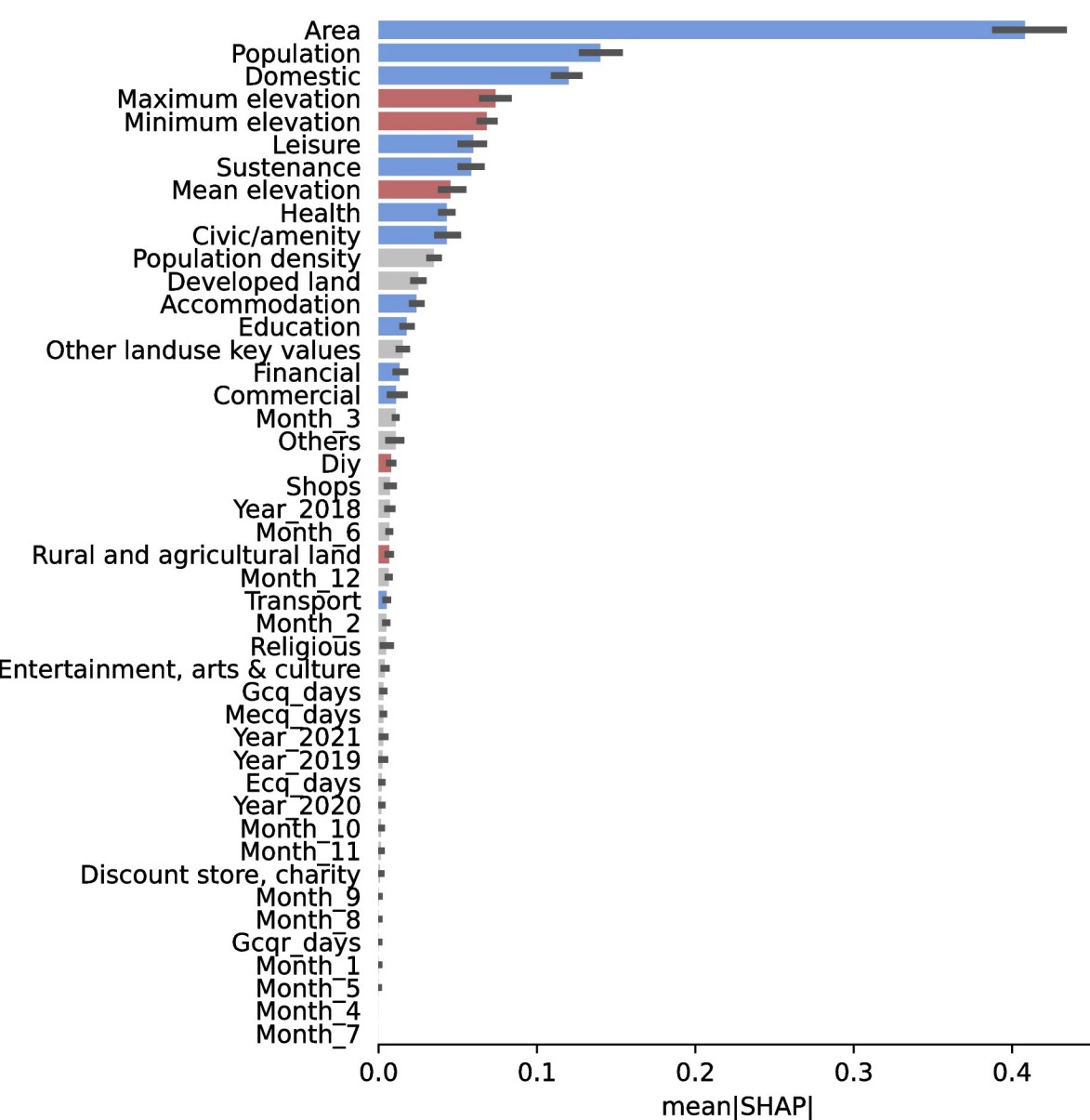

**Fig 8. SHAP Feature importance for the LightGBM implementation of a Gradient Boosting Tree.** Blue bars denote clear positive correlation, red for negative correlation, and gray bars have no clear correlation. Land area and population are the top two features that determine model predictions. Among amenities, *Leisure*, *Sustenance*, *Health*, and *Civic/Amenity*, are the top 4 predictive features, followed by *Developed land* and *Accommodation*.

pandemic. By adding the number of days for a quarantine type as additional features into our model, we incorporated the effects of the pandemic in our approach. The LightGBM implementation of Gradient Boosting was the best model with MAE and MAPE better by 16%, and 17% compared to base predictions. Although a slight improvement in MAE was observed, the negligible importance of pandemic-related features compared to amenities indicated that amenities are useful predictors that can capture long-term human mobility patterns. Amenities are sufficient in predicting water consumption even in the presence of disruptions, making the models robust and thereby diminishing the transient effects of quarantine protocols.

Finally, our models allowed us to identify amenity types that contribute most to prediction accuracy for our models. The top three amenity types were *Civic/Amenity*, *Leisure*, and *Sustenance*, but we also observed a decrease in the importance of *Leisure* coupled with an increase in importance for *Accommodation* during the pandemic. Our model and results show the roles various amenity types play in the overall consumption of cities and may provide helpful insights in the improvement of water consumption prediction.

## Supporting information

**S1 Appendix. Datasets.** Links to raw and processed datasets used in this work.
(PDF)

## Acknowledgments

The authors acknowledge the Manila Water Company, Inc. through Jon Michael H. Esteban for providing data used in this work. We also acknowledge John Peter Antonio, Michael Dorosan, Sebastian Ibañez, and Leodegario Lorenzo II for fruitful discussions.

## Author Contributions

**Conceptualization:** Damian Dailisan, Marissa Liponhay, Christopher Monterola.

**Data curation:** Damian Dailisan, Marissa Liponhay.

**Formal analysis:** Damian Dailisan, Marissa Liponhay, Christian Alis.

**Funding acquisition:** Christopher Monterola.

**Investigation:** Damian Dailisan, Marissa Liponhay.

**Methodology:** Damian Dailisan, Marissa Liponhay, Christian Alis, Christopher Monterola.

**Project administration:** Marissa Liponhay, Christopher Monterola.

**Resources:** Christian Alis.

**Software:** Damian Dailisan, Marissa Liponhay.

**Supervision:** Marissa Liponhay, Christopher Monterola.

**Validation:** Damian Dailisan, Marissa Liponhay, Christian Alis, Christopher Monterola.

**Visualization:** Damian Dailisan, Marissa Liponhay, Christian Alis, Christopher Monterola.

**Writing – original draft:** Damian Dailisan, Marissa Liponhay.

**Writing – review & editing:** Damian Dailisan, Marissa Liponhay, Christian Alis, Christopher Monterola.

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
