## [Decision Letter · Decision Letter 0]

14 Jan 2022

PONE-D-21-37246Amenity counts significantly improve water consumption predictionsPLOS ONE

Dear Dr. Dailisan,

Thank you for submitting your manuscript to PLOS ONE. After careful consideration, we feel that it has merit but does not fully meet PLOS ONE’s publication criteria as it currently stands. Therefore, we invite you to submit a revised version of the manuscript that addresses the points raised during the review process.

We look forward to receiving your revised manuscript.

Kind regards,

Shamsuddin Shahid

Academic Editor

PLOS ONE

Journal Requirements:

"The authors acknowledge the Manila Water Company, Inc. through Jon Michael H. Esteban for providing data used in this work. We also acknowledge John Peter Antonio, Michael Dorosan, Sebastian Iba˜nez, and Leodegario Lorenzo II for fruitful discussions. This research is primarily funded/monitored by the Department of Science and Technology (DOST) of the Philippines with Project No. 8419, under the CRADLE Program."

5. We note that Figure 1 in your submission contain map image which may be copyrighted. All PLOS content is published under the Creative Commons Attribution License (CC BY 4.0), which means that the manuscript, images, and Supporting Information files will be freely available online, and any third party is permitted to access, download, copy, distribute, and use these materials in any way, even commercially, with proper attribution. For these reasons, we cannot publish previously copyrighted maps or satellite images created using proprietary data, such as Google software (Google Maps, Street View, and Earth). For more information, see our copyright guidelines: http://journals.plos.org/plosone/s/licenses-and-copyright.

Reviewers' comments:

Reviewer's Responses to Questions

**Comments to the Author**

1. Is the manuscript technically sound, and do the data support the conclusions?

Reviewer #1: Yes

Reviewer #2: Yes

2. Has the statistical analysis been performed appropriately and rigorously? 

Reviewer #1: Yes

Reviewer #2: Yes

3. Have the authors made all data underlying the findings in their manuscript fully available?

Reviewer #1: Yes

Reviewer #2: No

4. Is the manuscript presented in an intelligible fashion and written in standard English?

Reviewer #1: Yes

Reviewer #2: Yes

5. Review Comments to the Author

Reviewer #1: The article in general is presented worth piece of work and essential components have been explained and elaborated. My comments are mainly technical rather conceptual.

The introduction section needs further elaboration on the main case study importance, available literature, and the selection of those ML models.

Case study and data description requires more essential information for better understanding of the readers.

The methodology section needs serious revision. Each models must be reported in more details. Modeling framework required flowchart elaboration. Mathematical concept must be there and add proper citation for the methodology section.

Figure 4 and 6 are importance and you need to further elaborate and discuss the attained results using the applied algorithm.

Reviewer #2: I think this manuscript is really interesting and it talks about important topic, which is the prediction of water consumption. But, I think it need some modifications before final acceptance to publish this manuscript.

1. In abstract, What is MAE?

2. please talk about Machine learning in introduction (1 paragraph)

3. please insert a link for each data you used in the body of the manuscript.

4. L65: what is the resolution of the population data

5. Please insert a spatial map contains the population data

6. Please use KGE with other metrics you used as it evaluate three statistical characteristics together (e.g., Pearson's correlation (r), spatial variability ratio and the normalized variance). You can see more details about it in:

"Hamed MM, Nashwan MS, Shahid S, Ismail T bin, Wang X, Dewan A, et al. Inconsistency in historical simulations and future projections of temperature and rainfall: A comparison of CMIP5 and CMIP6 models over Southeast Asia. Atmos Res. 2022;265: 105927. doi:10.1016/j.atmosres.2021.105927"

"Hamed MM, Nashwan MS, Shahid S. Inter-comparison of Historical Simulation and Future Projection of Rainfall and Temperature by CMIP5 and CMIP6 GCMs Over Egypt. Int J Climatol. 2022;n/a. doi:10.1002/joc.7468"

7. Please indicate the software you used in the analysis.

8. Please separate the results and discussion.

9. First paragraph in results and discussion should be in the introduction.

6. PLOS authors have the option to publish the peer review history of their article (what does this mean?). If published, this will include your full peer review and any attached files.

Reviewer #1: No

Reviewer #2: **Yes: **Mohammed Magdy Hamed

---

## [Author Response · Author response to Decision Letter 0]

23 Jan 2022

Please see attached rebuttal letter

---

## [Decision Letter · Decision Letter 1]

8 Mar 2022

Amenity counts significantly improve water consumption predictions

PONE-D-21-37246R1

Dear Dr. Dailisan,

We’re pleased to inform you that your manuscript has been judged scientifically suitable for publication and will be formally accepted for publication once it meets all outstanding technical requirements.

Kind regards,

Shamsuddin Shahid

Academic Editor

PLOS ONE

Additional Editor Comments (optional):

Reviewers' comments:

Reviewer's Responses to Questions

**Comments to the Author**

1. If the authors have adequately addressed your comments raised in a previous round of review and you feel that this manuscript is now acceptable for publication, you may indicate that here to bypass the “Comments to the Author” section, enter your conflict of interest statement in the “Confidential to Editor” section, and submit your "Accept" recommendation.

Reviewer #2: All comments have been addressed

2. Is the manuscript technically sound, and do the data support the conclusions?

Reviewer #2: Yes

3. Has the statistical analysis been performed appropriately and rigorously? 

Reviewer #2: Yes

4. Have the authors made all data underlying the findings in their manuscript fully available?

Reviewer #2: Yes

5. Is the manuscript presented in an intelligible fashion and written in standard English?

Reviewer #2: Yes

6. Review Comments to the Author

Reviewer #2: (No Response)

7. PLOS authors have the option to publish the peer review history of their article (what does this mean?). If published, this will include your full peer review and any attached files.

Reviewer #2: **Yes: **Mohammed Magdy Hamed

---

## [Editor Report · Acceptance letter]

10 Mar 2022

PONE-D-21-37246R1 

Amenity counts significantly improve water consumption predictions 

Dear Dr. Dailisan:

I'm pleased to inform you that your manuscript has been deemed suitable for publication in PLOS ONE. Congratulations! Your manuscript is now with our production department. 

Kind regards, 

on behalf of

Dr. Shamsuddin Shahid 

Academic Editor

PLOS ONE